# Characteristics of Neuroimaging and Behavioural Phenotype in Polish Patients with PIGV-CDG—An Observational Study and Literature Review

**DOI:** 10.3390/genes14061208

**Published:** 2023-05-31

**Authors:** Michal Hutny, Patryk Lipinski, Aleksandra Jezela-Stanek

**Affiliations:** 1Students’ Scientific Society, Department of Paediatric Neurology, Faculty of Medical Sciences in Katowice, Medical University of Silesia, 40-752 Katowice, Poland; michal.j.hutny@gmail.com; 2Department of Pediatrics, Nutrition and Metabolic Diseases, The Children’s Memorial Health Institute, 04-730 Warsaw, Poland; p.lipinski@ipczd.pl; 3Department of Genetics and Clinical Immunology, National Institute of Tuberculosis and Lung Diseases, 01-138 Warsaw, Poland

**Keywords:** *PIGV*, developmental delay, neuroimaging, behavioural disorders

## Abstract

Congenital disorders of glycosylation (CDGs) are a wide group of genetic diseases characterised by a severe clinical spectrum, consisting of developmental delays, dysmorphisms, and neurological deficits. Mutations in the *PIGV* gene lead to a disorder called hyperphosphatasia with impaired intellectual development syndrome 1 (HPMRS1), distinct from other CDGs in terms of hyperphosphatemia related to abnormal ALP activity and brachytelephalangy. This article discusses the phenotype of six Polish patients with HPMRS1 with a special focus on behavioural and imaging features, which were not addressed in 26 previously reported cases. The medical records of six patients aged 6 to 22 years were collected and analysed. In all cases, the same *PIGV* homozygotic mutation (c.1022C>A; p.Ala341Glu) was found, although the patients presented a diverse spectrum of neurological and developmental disorders, concerning in most cases the muscular tonus and general developmental delay. The most prevalent dysmorphic features included hypertelorism, high palate, and finger anomalies, whereas other characteristics present in all previously described cases, such as a short, broad nose and brachytelephalangy, were less frequently observed. Similarly to previous reports, the magnetic resonance (MR) and computed tomography (CT) head scans returned varied results, including physiological and pathological brain images in equal measure, the latter of which consisted of cortical atrophy, delayed myelination, hydrocephalus, and hypoplastic corpus callosum. Each patient exhibited symptoms characteristic of autism spectrum disorders, especially in terms of attention deficits, as well as controlling and expressing emotions. The most common type of sensory processing disorder was over-responsivity. Despite the low prevalence of HPMRS1, the patients reported in the literature presented a rather uniform phenotype, which does not correspond with the one found in each individual of the studied group. Behavioural disorders and sensory impairment require additional care and awareness considering the global developmental delay often observed in these patients.

## 1. Introduction

Post-translational protein modifications are a complex and varied group of processes, whose products are involved in a multitude of pathways and cycles [1]. One of these modifications is glycosylation with glycosylphosphatidylinositol (GPI), an anchoring molecule responsible for stabilising its over 150 target proteins in the cellular membrane [2]. The group of GPI-anchored proteins includes a variety of enzymes, receptors, and adhesion molecules [3]. GPI is synthesised in a multistage pathway, which involves the products of 23 genes [4]. Mutations altering the function of these genes lead to severe congenital syndromes known as GPI biosynthesis disorders (GPIBDs). Their phenotype consists of developmental delays (both motor and intellectual), encephalopathies, muscular hypotonia, a high prevalence of epileptic seizures, facial dysmorphisms, and cerebellar dysfunction. Although these disorders are considered rare, as in the case of some GPIBDs, the reports available in the literature are scarce; their actual prevalence might be higher. Despite the typical symptomatology resembling GPIBD phenotype, some patients cannot be included in groups of GPIBD patients for the purpose of analysis due to the lack of a necessary genetic assessment [5,6,7,8].

In 1970, Mabry et al. described for the first time a patient with severe cognitive impairment, seizures, and elevated serum alkalic phosphatase (ALP) activity. This syndrome was named hyperphosphatasia with impaired intellectual development syndrome 1 (HPMRS-1, # 239300) and linked to mutations in the *PIGV* gene. So far, a total number of six genes involved in GPI biosynthesis, including *PIGV*, *PIGO*, *PGAP2*, *PGAP3*, *PIGW,* and *PIGY*, have been reported to be responsible for HPMRS (namely HPMRS 1–6 orderly). HPMRS-1 is currently also known as GPIBD2 or *PIGV*-congenital disorder of glycosylation (PIGV-CDG), due to its difference in terms of phenotype from the typical clinical image of GPIBDs which derives from its alterations of alkalic phosphatase (ALP) activity and consecutive hyperphosphatemia.

To the best of the authors’ knowledge, to date, a group of 26 patients with HPMRS1 has been described in seven articles [9,10,11,12,13,14,15]. The aim of the study was to provide a report of six novel (unpublished) Polish patients affected with pathogenic variants in the *PIGV* gene, to further expand the clinical and molecular phenotype. The clinical features of a group from previously published studies were used for a comparison and discussion of similarities and differences. The issue which has not been addressed in the previous reports and was taken into consideration in the current study was the behavioural alterations observed in patients with *PIGV* mutations, which lay within the range of features typical for autism spectrum disorder (ASD), such as stereotypies, sensory processing disorders (SPD), and deficits in social interactions [16].

## 2. Materials and Methods

A total number of 6 patients, in whom biallelic pathogenic variants in the *PIGV* gene were found, were enrolled into the study group. Genomic DNA was extracted using an automated method (MagnaPure; Roche, CA, USA) from peripheral blood samples of the patients. The next-generation sequencing (NGS) of a targeted-gene panel, including all relevant genes associated with epilepsy, was used in two patients, whereas whole-exome sequencing (WES) was applied for the other four patients. The nomenclature of molecular variants followed the Human Genome Variation Society guidelines (HGVS, http://varnomen.hgvs.org/ (accessed on 20 April 2023) using a human cDNA sequence of the ABCB4 gene, which followed the Human Gene Mutation Database (HGMD, http://www.hgmd.cf.ac.uk (accessed on 20 April 2023).

The clinical, biochemical, and molecular data of patients were collected in an outpatient setting by the qualified specialists.

The study was conducted according to the guidelines of the Declaration of Helsinki and approved by the Children’s Memorial Health Institute Bioethical Committee, approval code: 36/KBE/2020, approval date: 28 October 2020, Warsaw, Poland.

PubMed, PubMed, Medline, and Online Mendelian Inheritance in Man (OMIM) databases were browsed and screened for previously published reports of patients with HPMRS1. Only the case studies presenting the exact genetic assessment, which confirmed the presence of pathogenic *PIGV* variants and, therefore, the diagnosis of Mabry syndrome itself were included for the purpose of analysis.

## 3. Results

The group consisted predominantly of male participants (5M; 1F), the majority of whom were children of school age (6–12 years), whereas the remaining two participants were young adults (17.5 and 22 years old, respectively) with a median age of 10.5 years.

All the patients carried the same homozygous pathogenic variant in the *PIGV* gene (NM_017837.4): c.1022C>A, p.(Ala341Glu). According to the American College of Medical Genetics and Genomics (ACMG) prediction of pathogenicity assessment, the homozygous variant c.1022C>A (p.Ala341Glu) in the *PIGV* gene fulfilled the following criteria: PP5, PP3, PM2, and PM5. It has been reported previously in ClinVar (Variation ID: 1284) and is very rarely found in gnomAD aggregated (AF: 0.0113%).

No exact relationship among the study group patients was known.

Their birth parameters were physiological in terms of weight, but the body length was abnormally high in two patients. Most of them, however, presented at this point with general hypotonia (5/6). The prenatal period was complicated in half of the patients, including ventricular system malformations (N = 2), polyhydramnios (N = 1), cholestasis in pregnancy (N = 1), renal malformations (N = 1), preterm birth (N = 1), and increased nuchal translucency (N = 1).

In further stages of life, all patients experienced global developmental delay. They exhibited intellectual disability ranging from mild (N = 3) and moderate (N = 3) to severe (N = 2), combined with delayed speech development, which, in some patients, remained persistent. Similarly, the patients achieved motor development milestones later than expected, whereas in some of them, their gait continued to be unstable and fine motor skills discoordinated. Despite the pathological EEG signal found in most patients (5/6), only two of them presented with epileptic seizures, one of whom developed refractory epilepsy.

All the patients presented with elevated ALP values as well as various signs and symptoms of gastrointestinal disorders (see Table 1). The other clinical findings included heart defects in two patients, as well as osteopenia/osteoporosis in one patient. The degree of persistent hyperphosphatasia varied between about 3.0 and 7.5 times the age-adjusted upper limit of the normal range in the studied patients.

### 3.1. Dysmorphic Features

In most cases, the general outlook of patients’ body shape was very slender, resulting from low body weight. Structural deformities were observed in chest morphology, such as pectus excavatum (2/6) and pectus carinatum (1/6), as well as in the extremities, where valgus, shoulder protraction, excessive knee extension, and flat foot were found. The physiological spinal curvatures were altered to both extremes—excessive lordosis and kyphosis. A wide range of other morphological abnormalities in relation to body regions are summarised below in Table 2.

The most frequent findings concerned facial features and the distal parts of extremities. More than half of the patients presented with hypertelorism of the eyes and gothic palate, as well as various finger anomalies, particularly clinodactyly and brachytelephalangy. It is worth noticing that macrocephaly was determined in three patients, in two of whom an enlarged ventricular system was previously found.

### 3.2. Neuroimaging Findings

All the patients were examined using magnetic resonance imaging (MRI) or computed tomography (CT) for the presence of any relevant pathologies in brain structures. Apart from individual hypoplastic changes in the cortex, corpus callosum, and olfactory bulb, the recurrent findings concerned an increase in cerebrospinal fluid (CSF) volume and dilation and asymmetry of the ventricular system, which together formed an image of hydrocephalus, as described below in Table 3.

### 3.3. Sensory Impairment

Every patient presented with either restricted visual (5/6) or auditory (4/6) acuity, the latter consisting predominantly of conductive hearing loss, successfully managed through ventilation drainage and tympanostomy tube insertion. Visual deficits in turn concerned various grades of hyperopia and astigmatism. The assessment of SPDs was conducted during the regular clinical examination in accordance with the Diagnostic Classification of Mental Health and Developmental Disorders of Infancy and Early Childhood (DC:0-3R)17.

On the basis of the above classification, the sensory disorders were divided into sensory modulation disorders (SMDs) comprising over/under-responsivity and sensory craving, sensory-based motor disorders (SBMDs) including dyspraxia and postural disorders, and sensory discrimination disorders (SDDs). Nearly every patient was found to present with a subtype of SDD, though most of the patients had tactile (3/6) and proprioceptive (2/6) deficits. Other SDD subtypes concerned a sense of balance; vestibular SDD was discovered in one patient.

Equally frequently found were SMDs, primarily in a form of over-responsivity to various stimuli, including touches to certain body parts, the consistency of materials, and sounds. Nevertheless, two patients simultaneously presented other subtypes of SMD. P2 tended to seek physical stimuli connected with the oral cavity by chewing and licking her toys, whereas P3—despite her over-responsivity to stimuli delivered to the head, face, hands, and feet—presented significant hyposensitivity in the upper extremities.

### 3.4. ASD Symptomatology

Although nearly all patients (5/6) displayed behavioural phenotypes typical for ASD, these features were not uniform in terms of presented disturbances and their onset. Interestingly, emotional problems (3/6), such as instability, inability to cope with changes, and reacting to them with violence, were found in patients whose everyday approach was rather joyful and friendly, and their interactions with the environment were sincere and forthright. Some patients had a tendency to develop echolalia, automatisms, stereotypic movements, and behaviour schemes. The difficulty which patients encountered during education were attention deficits, which significantly hindered their attempts at reading and writing.

### 3.5. Literature Review on Reported Patients

26 patients with pathogenic *PIGV* variants described in seven studies were included in this research. The genetic and clinical features of the reported patients are discussed below in the respective sections. Due to the lacking information concerning the pathogenic variants of the *PIGV* gene in the paper by Knaus et al. [9], the group of six newly introduced patients was not included in the discussion concerning the genetic features of previously reported HPMRS1 patients. The sensory impairment was not compared with the above group, as these issues were not described in any previous report of HPMRS1.

It is to be noted that reports whose clinical images were suggestive of the discussed disorder, including the original article by Mabry et al., were excluded from this analysis; nevertheless, this measure had to be taken for the sake of scientific preciseness.

## 4. Discussion

### 4.1. Genetic Diversity

An analysis of mutations causative for the condition of patients in the current study led to interesting observations. Despite the fact that the genotype of each patient was found to possess a homozygous c.1022C>A (p.Ala341Glu) variant, which is the most frequently found among reported GPIBD2 patients, the dysmorphic features of the current group were diverse and differed significantly from a rather homogenous profile of patients from the previous studies. The above differences were not apparent in equal measure in terms of clinical image, whose main components were similar between the studies. The variety of mutations found in both the present patients and the previous 20 patients, for whom the precise genetic data were available, is presented graphically below in Figure 1 [10,11,12,13,14,15].

### 4.2. Clinical Image

Subtle differences between the patients of the current study and those previously reported can be observed already during the neonatal period: the current patients were more frequently found to have an abnormal birth length (2/6 vs. 2/26) and less frequently occipitofrontal circumference (OFC) (3/6 vs. 4/6). Moreover, microcephaly, which was not found in any of the current patients, was present in two previous cases.

The biggest similarity between both groups concerned developmental delays. As in the previous reports, all patients described in this study, at some point in their life, experienced delays in every field of development—intellect, motor activity, and speech. Many of them could not compensate for these deficits, which, therefore, remained persistent throughout the rest of their life. A neurological symptom also present in both groups at an equal rate was muscular hypotonia (5/6 vs. 18/24); however, the current group had a varying degree of hypotonia—in some cases having a peripheral, whereas in the other, central, presentation.

The gastrointestinal presentations were a frequent finding in both groups, of which the most prominent symptom was constipation in both cases, although it differed significantly in pathogenesis. Many patients with HPMRS1 were diagnosed with Hirschsprung disease and consecutive megacolon (8/26), ileus, or anorectal malformations (9/26), which is naturally connected with the presence of constipation, whereas in the patients of the current study, none of the above structural findings were discovered. Although in one patient a colic atresia was present, no assessment of ganglion cells was conducted; therefore, the diagnosis of Hirschsprung disease cannot be made. Contrary to the lacking structural findings, the functional disorders were frequently met. In half of the patients, constipation was persistently observed; in two of them, it withdrew in childhood, and in the remaining patient it was episodic, altered by the episodes of diarrhoea.

Similarly in both groups, the cardiovascular pathologies were rather sporadic findings, in patients described in the literature consisting of atrial (2/26) and ventricular septal defects (2/26), as well as patent foramen ovale (1/26). The latter was also found in one patient of the current group, although the pathology was asymptomatic. During an echocardiogram, an increased myocardial thickness and hypertrabeculation of the left ventricle was found in another patient.

Increased activity of ALP is one of the key features of GPIBD2 and, as such, was a consistent finding in every patient of the current as well as previous groups.

### 4.3. Malformations

Patients with HPMRS1 described in the literature presented a rather uniform phenotype of malformations, which concerned, primarily, the facial gestalt and the morphology of fingers. The typical abnormalities included hypertelorism, up-slanting palpebral fissures, a short nose with a broad nasal bridge and tip, a tented upper lip vermilion, and brachytelephalangy (each present in 26/26 patients). Surprisingly, the above-mentioned features were not observed in the current study population. Of all the above characteristics, the only frequently repeating one was apparent hypertelorism (4/6); the others were rare or even nonexistent.

In the current study, as stated above, the most prominent malformations included hypertelorism, but also a high palate (4/6), macrocephaly (3/6), a high forehead (2/6), a broad nasal tip (2/6), a narrow mouth (2/6), pectus excavatum (2/6), and hypoplastic fingernails (2/6). Some of these features were also present in previous cases, such as nasal, oral, and pectoral anomalies, but palate abnormalities and macrocephaly were significantly less frequent in previous reports. Although finger anomalies were found in most patients of the current study, brachytelephalangy itself was present in only two of them. The other presentation of finger dysmorphism was clinodactyly of the fifth finger.

Interestingly, abnormalities of the urogenital system, which were a frequently repeating feature among the patients in previous reports of GPIBD2 cases, were absent in the group of the current study.

### 4.4. Neuroimaging

Pathological findings in brain imaging were not frequent in either of the groups; nonetheless, the group described in the current study showed significantly more of these alterations. A structure which was altered to the highest extent was the ventricular system of the brain, which was found to be abnormal in half of the patients, either in the form of hydrocephalus or as an increased CSF volume. The latter finding was observed previously in one HPMRS1 patient. Other abnormalities, such as hypoplasia of the corpus callosum, olfactory bulb, or cerebellar vermis, were sporadic, appearing only in individual patients. Delayed myelination and corpus callosum hypoplasia were the only neuroimaging findings which repeated between the current and the previously described patients.

### 4.5. Sensory Impairment and Behavioural Phenotype

An issue which has been, up to this point, left unnoticed in most of the available sources is the behavioural disorders of patients with Mabry syndrome which lay within the spectrum of ASD. A single previous HPMRS1 study addressed the issue of autism, stating a lack of ASD symptoms in the reported group [11]. Most patients of the current study presented features of autism (4/6), such as schematic behaviour, stereotypic movements, echolalia obsessions, and selective detail memorisation [16,17,18]. One of the criteria for the diagnosis of ASD is disorders of social interactions, which were also present in the studied group—the patients were often unable to process the changes and reacted emotionally inadequately to the situation [16,18]. Attention deficits were also observed in GPIBD2 patients who encountered problems in focusing on given tasks, which influenced their education.

Patients with HPMRS1 were rarely reported to present any sensory deficits. In two of these cases, an optic atrophy was found, resulting in impaired visual acuity, whereas hearing was affected in one patient due to recurrent ear infections, which led to conductive hearing loss.

The above observations differ significantly from the clinical image of the group in the current study, as all patients suffered from either auditory or visual impairment, or as in the case of three patients, from both. Similarly to previous patients, hearing loss in the current group resulted from the abundance of fluid in the ear cavities and, therefore, was treated with tympanopuncture and drainage.

None of the previous reports concentrated on SPDs present in patients with Mabry syndrome; therefore, this issue may be addressed solely on the basis of the current findings. This type of disorder is frequently found in children with ASD, which corresponds to the prevalence of autistic traits among the studied patients. Furthermore, SMDs, a subtype of SPDs, are present in over 90% of children with ASD [19]. Taking the above into consideration, it is rather expected that nearly every patient in the current group presented with an SMD, either in the form of over-responsivity (5/6) or under-responsivity (1/6).

Concerning SDDs, an increased sensitivity to auditory stimuli is widely described in individuals with ASD, surpassing in these terms even visual receptive alterations, which were initially considered the most affected compared with the healthy population [20,21]. On the contrary, patients of the current group presented the most frequently with SDDs associated with touch—tactile (3/6) and proprioceptive (2/6) SDDs. Only one patient presented with hyperacusis to sounds, whereas no visual SDDs were observed. The results of the previous studies suggest that tactile hypersensitivity might be associated with altered γ-aminobutyric acid (GABA) neurotransmission [22]. The above findings draw attention to the assessment of this type of neuronal activity in HPMRS1 patients, as it might provide a useful insight into the atypical phenotype of SPD in these patients, as well as into the mechanisms underlying the condition which limits, to some extent, their functioning in everyday life.

### 4.6. Limitations

An obvious limitation of the study is the small group size, which, considering the low prevalence of GPIBD2, cannot be omitted. The analysis and comparison of phenotypes between the current group and patients described in the available literature were restricted due to the availability of data. The varied approaches to the significance of different features of patients’ clinical image restricted an adequate discussion on this topic. As stated above, some cases were excluded from the analysis due to the lack of a genetic assessment necessary for the diagnosis of HPMRS1, despite rather suggestive symptomatology.

## 5. Conclusions

The current study introduces six new Polish patients diagnosed with Mabry syndrome who, despite possessing the most frequently found pathogenic mutation of the *PIGV* gene (c.1022C>A, p.(Ala341Glu)), presented with a phenotype very distinct from previously reported cases. Although, naturally, the current patients shared traits such as general motor development, high ALP activity, hypotonia, and gastrointestinal disorders with other HPMRS1 patients, many other features considered typical for GPIBD2 were significantly less frequent or even absent in the population of this study. The most prominent differences concerned the dysmorphisms of the face and extremities. To the best of the authors’ knowledge, this is the first study to address the issue of behavioural disorders and sensory impairment in patients with HPMRS1, which proved to be frequent among this population and, as such, requires adequate therapeutic strategies.

## Figures and Tables

**Figure 1 genes-14-01208-f001:**
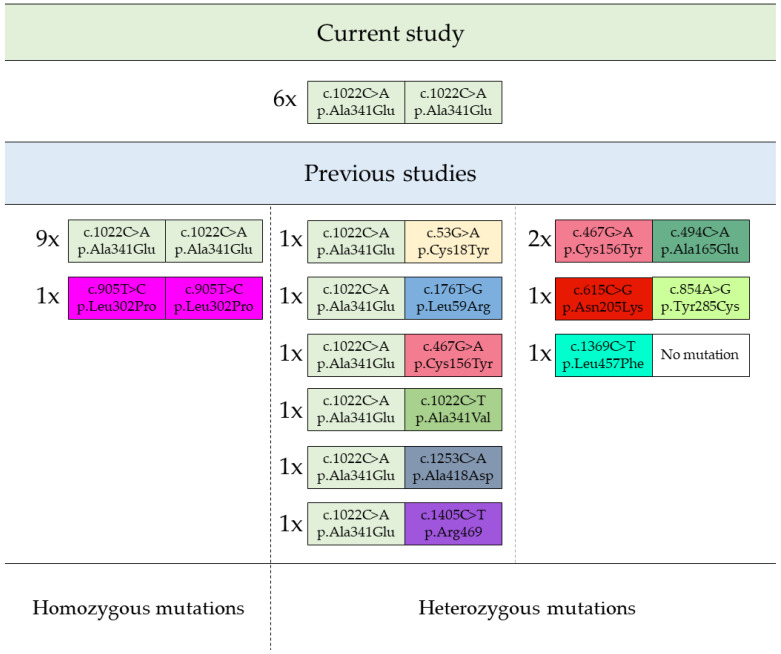
Homo- and heterozygous mutations of *PIGV* gene found in GPIBD2 patients. Footnote: the quantity of patients harbouring given *PIGV* mutations are displayed next to each representative variant as “number of patients; x; variant”.

**Table 1 genes-14-01208-t001:** The summary of clinical findings in patients of the current study.

Symptom	P1	P2	P3	P4	P5	P6
Constipation	+	+	+	+	+	+
Diarrhoea	+	−	−	−	−	−
Tympany	−	+	−	+	−	−
Hyperphagia	−	−	−	+	−	−
Intestinal inflammation	+	−	−	−	−	−
Gastritis	+	−	−	−	−	−
Colic atresia	−	+	−	−	−	−
Splenomegaly	−	−	−	+	+	−
Nutritional neophobia	−	−	−	−	−	+
Heart defects	−	+	−	−	+	−
ALP at given age (IU/L)	2Y: 200022Y: 700–900	17.5Y: 820	3Y: 5006Y: 979	6.5Y: 1011	9Y: 696–2000	12Y: 1006

**Table 2 genes-14-01208-t002:** Dysmorphisms in patients in relation to respective body regions.

Dysmorphic Feature	P1	P2	P3	P4	P5	P6	Total
Head and face							
Macrocephaly	+	−	+	−	+	−	3/6
Plagiocephaly	−	−	−	−	+	−	1/6
High forehead	−	−	−	+	+	−	2/6
Ears							
Large, fleshy ear lobes	+	−	−	−	−	−	1/6
Free ear lobes	−	−	+	−	−	−	1/6
Eyes							
Broad eyebrows	+	−	−	−	−	−	1/6
Hypertelorism	+	+	−	+	+	−	4/6
Ptosis	−	−	+	−	−	−	1/6
Epicanthus	+	−	−	−	−	−	1/6
Nose							
Broad nasal bridge	−	−	−	+	−	−	1/6
Broad nasal tip	−	−	+	+	−	−	2/6
Mouth and oral cavity							
Gothic palate	+	−	+	+	−	+	4/6
Cleft palate	−	+	−	−	−	−	1/6
Tented upper lip vermilion	+	−	−	−	−	−	1/6
Down-slanting lips	−	+	−	−	−	−	1/6
Narrow mouth	−	+	+	−	−	−	2/6
Open bite	+	−	−	−	−	−	1/6
Retrognathism	−	−	+	−	−	−	1/6
Extremities							
Pectus excavatum	−	−	+	−	−	+	2/6
Pectus carinatum	−	−	−	+	−	−	1/6
Short extremities	−	−	−	−	+	−	1/6
Large feet	−	−	−	−	+	−	1/6
Finger anomalies	+	−	+	+	−	+	4/6
Brachytelephalangy	+	−	+	−	−	−	2/6
Hypoplastic fingernails	−	−	+	−	+	−	2/6

**Table 3 genes-14-01208-t003:** Brain malformations found in the study group.

Imaging Findings	P1	P2	P3	P4	P5	P6	Total
Cortical atrophy	+	−	−	−	−	−	1/6
Hydrocephalus	+	−	−	+	+	−	3/6
Increased CSF volume	+	−	+	−	+	−	3/6
Ventricular asymmetry	−	−	+	−	−	−	1/6
Corpus callosum hypoplasia	−	+	−	−	−	−	1/6
Delayed myelination	−	+	−	−	−	−	1/6
Pineal gland cyst	−	−	−	−	+	−	1/6
Olfactory bulb hypoplasia	−	−	−	−	+	−	1/6

## Data Availability

All data generated or analysed during this study are included in this published article.

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
