# Peer review of "Characteristics of Neuroimaging and Behavioural Phenotype in Polish Patients with PIGV-CDG—An Observational Study and Literature Review"

_genes, 2023, doi:10.3390/genes14061208_

Round 1
Reviewer 1 Report
1. Despite being described previously in other reports in the literature, I suggest authors to include the ACMG prediction of pathogenicity assessment in their text. In this case, the homozygous variant c.1022C>A (p.Ala341Glu) (NM_017837.4) in the PIGV gene fulfilled the following criteria: PP5, PP3, PM2, and PM5. It has been reported previously in ClinVar (Variation ID: 1284) and is very rarely found in gnomAD aggregated (AF: 0.0113%).
2. I suggest the correction of reference 17 (Zeanah et al., 2016) which has been described only with capital letters.
3. I suggest authors to include a Figure with the main neuroimaging findings observed.
Author Response
Dear Reviewer,
We are very grateful for Your comments.
1. Despite being described previously in other reports in the literature, I suggest authors to include the ACMG prediction of pathogenicity assessment in their text. In this case, the homozygous variant c.1022C>A (p.Ala341Glu) (NM_017837.4) in the PIGV gene fulfilled the following criteria: PP5, PP3, PM2, and PM5. It has been reported previously in ClinVar (Variation ID: 1284) and is very rarely found in gnomAD aggregated (AF: 0.0113%).
Answer: It was corrected as advised. Please, see the revised version of manuscript with corrections made in red.
2. I suggest the correction of reference 17 (Zeanah et al., 2016) which has been described only with capital letters.
Answer: It was corrected as advised. Please, see the revised version of manuscript with corrections made in red.
3. I suggest authors to include a Figure with the main neuroimaging findings observed.
Answer: Unfortunately, we do not have CT/MR pictures to be published.
Reviewer 2 Report
In the present manuscript entitled “Characteristics of neuroimaging and behavioural phenotype in Polish patients with PIGV-CDG – an observational study and literature review”, the authors reported six Polish patients with HPMRS1 carrying the mutation in the PIGV gene, and conducted a comparative assessment of their clinical as well as genetic characteristics with the cases that had previously been reported. Although clinical data are well-described and discussed in the manuscript, there are still several concerns that must be addressed.
1. Although the authors briefly describe Mabry syndrome in the sections of Materials & Methods and Discussion, they never do in the section of Introduction. They should explain Mabry syndrome as well as HPMRS1 in Introduction, as it is not clear for the readers, especially those who are not familiar to this field, as to whether these diseases are the same or the different one.
2. The authors must need to provide us with how the sequence information was obtained; e.g. the patient sample, DNA extraction methods, DNA sequencing methods and analysis. The description as “The clinical, biochemical and molecular data of patients were collected in an outpatients setting by the qualified specialists” in the manuscript is not enough.
3. The authors reported that six Polish patients had the same mutation in the PIGV gene; c.1022C>A; p.A341E. Are there any kin relationship among patients. If patients belong to the same relative members and/or family, the accounting number of patients as “six” is inappropriate in the genetic point of views.
4. The authors should present the representative image data of MRI and CT.
5. Figure 1 is hard to understand. The reviewer could not understand what the authors intended to show by this figure. At least, the authors must clearly describe six newly identified and 26 previously reported cases.
Author Response
Dear Reviewer,
We are very grateful for Your comments.
Although the authors briefly describe Mabry syndrome in the sections of Materials & Methods and Discussion, they never do in the section of Introduction. They should explain Mabry syndrome as well as HPMRS1 in Introduction, as it is not clear for the readers, especially those who are not familiar to this field, as to whether these diseases are the same or the different one.
Answer: It was corrected as advised. Please, see the revised version of manuscript with corrections made in red.
The authors must need to provide us with how the sequence information was obtained; e.g. the patient sample, DNA extraction methods, DNA sequencing methods and analysis. The description as “The clinical, biochemical and molecular data of patients were collected in an outpatients setting by the qualified specialists” in the manuscript is not enough.
Answer: It was corrected as advised. Please, see the revised version of manuscript with corrections made in red.
The authors reported that six Polish patients had the same mutation in the PIGV gene; c.1022C>A; p.A341E. Are there any kin relationship among patients. If patients belong to the same relative members and/or family, the accounting number of patients as “six” is inappropriate in the genetic point of views.
Answer: It was corrected as advised. Please, see the revised version of manuscript with corrections made in red.
The authors should present the representative image data of MRI and CT.
Unfortunately, we do not have CT/MR pictures to be published.
Figure 1 is hard to understand. The reviewer could not understand what the authors intended to show by this figure. At least, the authors must clearly describe six newly identified and 26 previously reported cases.
Answer: It was corrected as advised. A new Figure was created and uploaded in the revised version of manuscript.
Round 2
Reviewer 2 Report
In the revise manuscript entitled “Characteristics of neuroimaging and behavioural phenotype in Polish patients with PIGV-CDG – an observational study and literature review”, the authors most appropriately improved the context of the manuscript by responding to my original concerns and comments, there are still few minor concerns that must be addressed.
1. Figure 1: It is still hard to understand, and its legend is also poorly described. For example, what does mean by the description as “?x”, “1x”, and 2x”?
2. In this figure, 17 different variants are shown. However, as described in line 180, the authors analyzed 26 patients with pathogenic PIGV variants. How many variants were indeed reported in 7 previous reports in total? The authors must consistently and explicitly show the precise number of patients and variants both in the figure and text by showing the genotypes of all 26 patients.
Author Response
Dear Reviewer,
Thank You so much for Your comments.
The Figure was corrected as advised. Please, see the revised version.